# Are Gastrointestinal Microorganisms Involved in the Onset and Development of Amyloid Neurodegenerative Diseases?

**Vladimir I. Muronetz** [1,2,*], **Lidia P. Kurochkina** [1], **Evgeniia V. Leisi** [3] and **Sofia S. Kudryavtseva** [1]

1   Belozersky Research Institute of Physico-Chemical Biology, Lomonosov Moscow State University, Leninskie Gory 1, Bld 40, 119991 Moscow, Russia; lpk@belozersky.msu.ru (L.P.K.); sofiia.kudriavtceva@gmail.com (S.S.K.)
2   Butlerov Chemical Institute, Kazan Federal University, Kremlevskaya 18, 420008 Kazan, Russia
3   Faculty of Bioengineering and Bioinformatics, Lomonosov Moscow State University, Leninskie Gory 1, Bld 73, 119991 Moscow, Russia
*   Correspondence: vimuronets@belozersky.msu.ru

**Abstract:** This review discusses a few examples of specific mechanisms mediating the contribution of the GIT microbiota to the development of amyloid neurodegenerative diseases caused by the pathologic transformation of prion protein, or alpha-synuclein. The effect of the bacterial GroE chaperonin system and phage chaperonins (single-ring OBP and double-ring EL) on prion protein transformation has been described. A number of studies have shown that chaperonins stimulate the formation of cytotoxic amyloid forms of prion protein in an ATP-dependent manner. Moreover, it was found that *E. coli* cell lysates have a similar effect on prion protein, and the efficiency of amyloid transformation correlates with the content of GroE in cells. Data on the influence of some metabolites synthesized by gut microorganisms on the onset of synucleinopathies, such as Parkinson's disease, is provided. In particular, the induction of amyloid transformation of alpha-synuclein from intestinal epithelial cells with subsequent prion-like formation of its pathologic forms in nervous tissues featuring microbiota metabolites is described. Possible mechanisms of microbiota influence on the occurrence and development of amyloid neurodegenerative diseases are considered.

**Keywords:** neurodegenerative disease; chaperones; gut microbiota; prion protein; alpha-synuclein



## 1. Introduction

The study of possible relationships between microbiota and the development of amyloid neurodegenerative diseases has become a trend in science over the last decade. Most of the works concern the correlation between the occurrence of neurodegenerative diseases and microbiota features. For example, there is quite a lot of research on the role of the microbiota in the development of Alzheimer's disease (AD). Several reviews devoted to this problem have declared that Alzheimer's patients have a deregulated microbiota [1] and that gut microbial metabolites, such as pro-inflammatory factors and short-chain fatty acids, could affect the pathogenesis of AD due to synaptic dysfunction and neuroinflammation, which contribute to cognitive decline [2,3]. However, the specific mechanisms of such an interrelation are still relatively unknown. In this review, we will address only two amyloid neurodegenerative diseases—prion diseases and Parkinson's disease—and summarize the information on the possible influence of microorganisms from the gastrointestinal tract of humans and animals on their manifestation. Special attention will be paid to the role of bacterial and bacteriophage chaperonins in the amyloid transformation of prion protein and alpha-synuclein, which is the cause of pathological changes in nervous tissues in these diseases.

## 2. Basic Properties of Prion Protein and Peculiarities of Oral Transmission of Its Infectious Forms

Prion protein (PrP) is a conservative protein that consists of 253 amino acids and is localized at the outer layer of the neuron plasma membrane [4]. There are two domains in its tertiary structure: the unstructured N-terminal and the globular C-terminal, containing three α-helices and two β-sheets [5–7]. The normal cellular form of PrP (PrP$^c$) is able to transform into an infectious scrapie isoform (PrP$^{Sc}$) by structural reorganization of the globular C-domain (Figure 1) [8]. As recently shown, PrP$^{Sc}$ is characterized by fibril formation with a parallel in-register intermolecular β-sheet (PIRIBS) architecture based on a regular cross-β structure in which PrP molecules are aligned parallel in-register to each other [9].

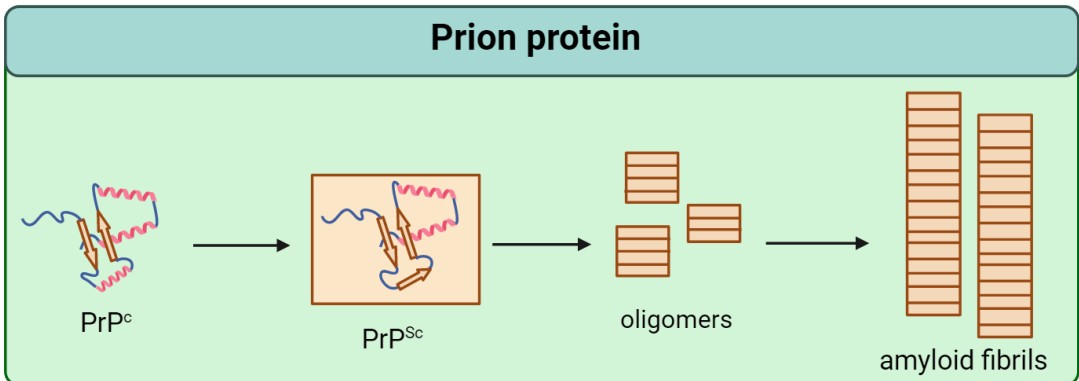

**Figure 1.** Scheme of the pathological transformation of prion protein (PrP). Single arrows are dedicated to irreversible change of structure.

Prion infection of the organism occurs when a scrapie isoform of prion protein enters the gastrointestinal tract (GIT) [10,11]. However, the mechanisms involved in the passage of PrP$^{Sc}$ from the GIT to nervous tissues, where the infectious particles induce amyloid transformation of the cellular form of prion protein, are unknown and little discussed in the specialized literature [9]. It is stated that the remarkable resistance of the PrP scrapie isoform to the action of proteinases allows it to be retained in all parts of the GIT without reducing infectivity [12–14]. Another aspect that may be related to the involvement of microbiota in the onset of prion infection concerns the peculiarities of interspecies transmission of PrP$^{Sc}$.

It is known that when prion protein particles arising from Kuru disease ingest another human, the probability of disease occurrence is very high [15,16]. A similar pattern is seen in cases of intraspecific transmission of the transmissible spongiform encephalopathies within other mammals—cattle (BSE), sheep (scrapie), etc. Simultaneously, cross-species transmission of infectious prion is difficult (e.g., humans contracting BSE) or impossible (there are no known cases of human infection with the sheep form—scrapie) [17,18]. The reasons why interspecies transmission occurs in some cases and not in others are unknown, but there is a possibility that it is determined by the composition of the microbiota, which could define the efficiency of transport from the gastrointestinal tract to nervous tissues and, therefore, the probability level of PrP pathologic transformation.

## 3. The Role of Chaperones in the Pathological Transformation of Prion Protein and Involvement of Microbiota in This Process

The processes of different proteins' translation, their proper folding, maturation, and transport take place in the endoplasmic reticulum (ER) with the participation of chaperones [19,20]. In mammals, prion protein formation occurs by the same mechanism. Then PrP is exposed to the membrane of neurons, forming the diffuse neuroendocrine system and lymphoreticular cells [21]. Prion protein can also be found in the cytosol of neurons [22,23].

Disruption of protein folding in the cell leads to the accumulation of improperly folded proteins in the ER, resulting in endoplasmic reticulum stress (ER stress) [24]. Since prion diseases are also associated with the accumulation of protein amyloid aggregates in the cell, it is not surprising that ER stress is frequently observed in prion disease models and contributes significantly to the development of these pathologies [25–27]. ER stress has also been observed in patients with sporadic and variant forms of Creutzfeldt-Jakob disease [28]. In addition, one of the significant contributors to ER stress is the eukaryotic chaperone Grp78 [29], a member of the Hsp70 family [30]. It is one of the most abundant proteins in the ER, making it an important factor in protein folding within cells [31]. Grp78 interacts with mutant PrP and mediates its degradation by the proteasome, suggesting that Grp78 accompanies the folding of the prion protein during its de novo synthesis [32–34]. Grp78 levels were increased in neuroblastoma cells infected with the scrapie isoform of PrP [28,35], and in mice infected with prion protein [36]. More importantly, elevated levels of this particular chaperone have been found in brain samples from patients with sporadic Creutzfeldt-Jakob disease [28]. The chaperone Grp78 appears to be involved in the folding of the native prion protein. Therefore, the accumulation of mutant or infectious forms of PrP may sequester Grp78 functioning and cause ER stress with stimulation of Grp78 synthesis, which in turn could lead to a worsening of the situation. It has also been shown that chaperone levels are altered in prion-infected mice and that the response vector depends on the prion strain. Levels of Hsp60, Hsp70, Hsp90, Grp78, and Grp94 were increased in the ME7 and 87V strains, whereas in the 22L and 139A strains, Hsp60 levels were decreased, and the increase in Hsp70 and Hsp90 levels was less pronounced [37]. The chaperones Hsp72 and Hsp73 have also been implicated in prion diseases. The brains of scrapie-infected mice have been found to contain an abnormally high number of lysosomes enriched in PrP and Hsp73 [38]. Hsp72 levels increase in neurodegenerative diseases [39], as well as in cellular [40] and animal models of prion diseases [41].

Looking at the described issue from another angle, the molecular chaperone Hsp70 plays a protective role in prion diseases. It can directly interact with PrP and prevent the accumulation of its misfolded isoforms [42–44]. Furthermore, induction of Hsp70 by pharmacological agents helps to reduce the accumulation of prion protein pathological forms, promotes their degradation, and improves patients' motor abilities [45,46]. In addition, some chaperones may promote the degradation and excretion of PrP pathological forms. In the case of mammals, the Hsp70/DnaJ-1/Hsp110 disaggregation system fulfills this role. At the same time, overexpression of DnaJ-1 and Hsp110 also reduces the toxic effects of prion in flies [47].

In general, chaperones play an important role in the implementation of PrP's natural functions, participating in its folding and transport to neuronal membranes. However, it should be especially noted that the molecules of the prion protein also influence the chaperone system. In fact, mutant or infectious forms of PrP can block chaperones and thus slow down their normal functioning in relation to other proteins. For example, the eukaryotic chaperonin TRiC/CCT promotes the amyloid transformation of PrP monomers and induces further growth of its oligomers while not being able to interact with denatured molecules of its substrate protein [48].

The chaperones Hsp60 and Hsp104 have also been found to stimulate the pathological transformation of prion protein [49]. The role of Hsp60 in the emergence of PrP infectious forms has been extensively studied. It has been reported that chaperonins of the Hsp60 family are able to stimulate PrP aggregation regardless of the organism in which the chaperonin occurs [50]. In particular, the bacterial complex of chaperonins GroEL-GroES (GroE) promoted amyloid aggregation of prion protein in an ATP-dependent manner in in vitro models [51,52]. The monomeric form of PrP binds to the apical domain of GroEL, presumably leading to its conversion and subsequent formation of amyloid structures, as shown by cryogenic electron microscopy and molecular modeling [53].

It was later found that incubation of recombinant sheep prion protein in the presence of the bacterial complex GroE resulted in the formation of spherical aggregates of relatively

small size, about 200 nm. It was suggested that such spherical particles, which probably contained amyloid forms of prion protein, could pass through the walls of the gastrointestinal tract, penetrate into nervous tissue, and cause the formation of amyloid structures by interacting with cellular PrP. It was also shown that incubation of recombinant sheep prion protein with *E. coli* cell lysates also stimulates its amyloid transformation. *E. coli* cell lysates with GroE overproduction cause more pronounced amyloid transformation, whereas strains without GroE have a minimal effect on prion protein [54]. Thus, it was shown that bacterial chaperones released during the lysis of intestinal microorganisms could be involved both in the amyloid transformation of prion protein and in the realization of mechanisms responsible for the transport of its aggregates to nervous tissues through the formation of prion particles of small size (Figure 2).

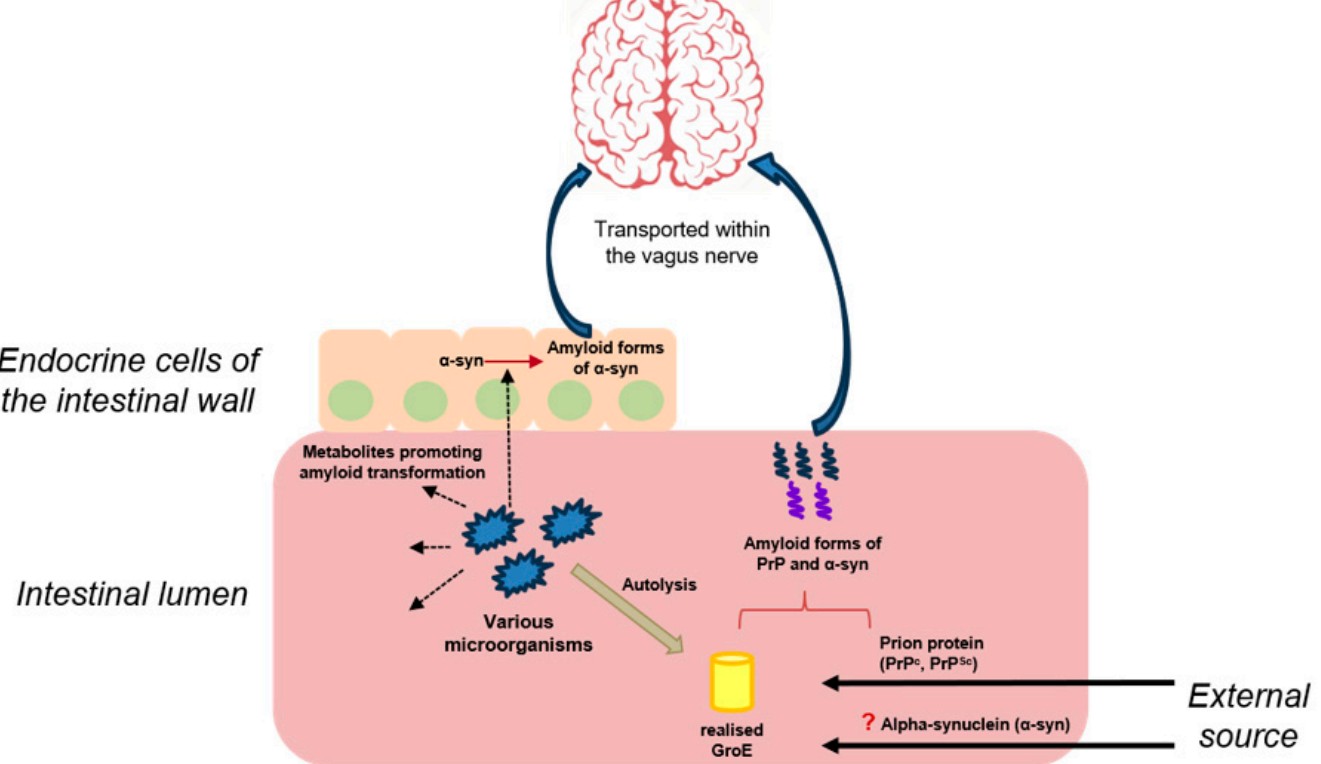

**Figure 2.** The presumed contribution of the microbiota to the development of prion diseases and synucleinopathies. Dashed arrows are dedicated to the release of metabolites by microorganisms' cells; red arrow/parenthesis—to the amyloid transformation of different proteins; bold arrows—to the external input; "?"—to a possible but not proven route for alpha-synuclein to enter the body; blue arrows—to the spreading pathway of amyloid forms of proteins.

In addition, according to literature data, the process of microbial colonization of mice with a sterile intestine initiates signaling mechanisms affecting neural circuits involved in motor control and anxiety behavior [55]. The gut microbiota was also found to influence the balance between pro- and anti-inflammatory immune responses during experimental autoimmune encephalomyelitis. These observations suggest a link between the microbiota and the development of extraintestinal inflammatory diseases such as multiple sclerosis [56].

Two GroEL-like phage chaperonins have a similar effect on prion protein transformation. Single-ring OBP and double-ring EL have also been shown to stimulate PrP fibrillation in an ATP-dependent manner. In the presence of ATP, chaperonins initiate the conversion of prion protein monomers into short amyloid fibrils, with their further aggregation into larger particles with reduced cytotoxicity. At the same time, fibrils formed by phage chaperonins

differ in morphology and properties from fibrils formed spontaneously from monomeric PrP in the presence of denaturants at an acidic pH [57].

It's also worth mentioning that a major contribution to the understanding of the mechanisms underlying the influence of chaperones on amyloidogenesis has been made by studying these processes in yeast, which possesses its own prion protein Sup35 and a set of chaperones [58]. Although yeast models are only remotely relevant to the occurrence of prion diseases in animals, such models are conveniently used to identify general patterns of chaperone effects on the amyloid transformation of proteins [59,60]. In addition, the expression of mammalian amyloid proteins in yeast cells allows these models to be approximated by those based on mammalian cells [61]. In the case of yeast, overexpression of the chaperone Hsp104 was found to promote the disaggregation of yeast prion Sup35 aggregates [62]. Further in vitro experiments showed that Hsp104 at high concentrations does indeed bind amyloidogenic oligomers and disassemble Sup35 fibrils, thereby preventing the propagation of the [PSI+] phenotype. However, at low concentrations, Hsp104 catalyzes the assembly of oligomeric intermediates and fragment fibrils [63]. The chaperone Hsp104 also promotes the formation of an infectious form of another yeast prion, Ure2 [64].

An unusual relationship between GroEL-like chaperonins (Hsp60) and the cellular form of prion protein was discovered while studying the mechanisms of human infection with *Brucella abortus* bacteria. Prion protein is expressed in significant amounts on the surface of human intestinal M-cells, whereas Hsp60 is expressed on the surface of *Brucella abortus* cells. It is hypothesized that in the case of oral infection, *Brucella abortus* could enter host cells via M-cells, using PrP$^c$ as a receptor that binds Hsp60 [65,66]. In addition to M-cells, a similar mechanism facilitates *Brucella* entry into macrophages, which also express PrP$^c$ [67].

## 4. The Role of Microbiota in the Occurrence of Prion Diseases

Despite the interest, there have been few studies conducted to determine the microbiota's role in the development of prion diseases. However, it is known that *Akkermansia* and *Lachnospiraceae* taxa are increased in deer (*Odocoileus virginianus*) with chronic wasting disease [68]. It is also interesting that mice infected with the RML strain had an altered microbiota composition. Specifically, they had increased *Proteobacteria* and a reduced level of *Saccharibacteria*. At the family level, *Lactobacillaceae* and *Helicobacteraceae* counts increased, while *Prevotellaceae* and *Ruminococcaceae* decreased. *Proteobacteria* growth is a symptom of dysbacteriosis and also plays a role in causing intestinal inflammation. At the same time, the reduction in *Prevotellaceae* levels may be associated with amplified intestinal permeability. Simultaneously with changes in microbiota composition, a significant alteration of the metabolome in the GIT has also been identified. This may also contribute to the development of neurodegenerative diseases [69].

Although it is evident that the gut microbiome is altered during the development of prion diseases, it remains uncertain whether the microbiota directly affects the development of such diseases. As mentioned above, the scrapie isoform of PrP is known to be highly stable and retains its infectivity during digestion [70,71], thereby surviving passage through the GIT [72,73]. Entering the intestine, prion particles are likely to be internalized by M-cells and spread through lymphoid tissues and the peripheral nervous system, from where they could enter the CNS [74]. M-cells are specialized epithelial cells that bind antigens and transport them to lymphocytes. The number of M-cells can be influenced by intestinal microbes and inflammation. For example, when Salmonella levels increase, M-cell numbers increase [69]. Increased M-cell numbers in turn increase the uptake of prions from the intestinal lumen, increasing susceptibility to prion diseases in mice [75]. Thus, it is possible that disruption of the gut microbiota may increase susceptibility to prion diseases.

Alternatively, several studies show that the microbiota controls microglia development and function in the CNS. Furthermore, microglia disorders have also resulted from the temporary destruction of the host microbiota and the limitation of its complexity. The gut microbiota prepares the brain for the rapid development of immune responses by

controlling the immune function of microglia [76]. Meanwhile, one of the first pathological features observed in the prion-infected CNS is the activation of microglia. This is important for neuroprotection, as microglia are capable of engulfing apoptotic cells. Moreover, prion infection is accelerated in microglia-deficient mice, whereas microglia defects with impaired immune responses have been observed in germ-free mice. Therefore, it is possible that altering the microbiota composition during antibiotic treatment or prion infections may accelerate disease development [77].

However, there are some hard-to-explain contradictions between studies that may shed light on this theory. Early experiments showed that germ-free mice injected intracerebrally with the Chandler mouse-adapted scrapie isolate demonstrated a longer survival time than control mice [78]. Later, a longer survival time of sterile mice infected with the ME7 prion strain was observed only after intraperitoneal inoculation [79]. In addition, a recent study showed that the absence of the commensal microbiota did not affect the development of prion infection in mice after intraperitoneal and intracerebral inoculation with the 22L strain [80].

Considering this information, it is challenging to reach a definitive conclusion about the impact of the microbiota on prion infection development. However, it cannot be completely ruled out that the microbiota plays some role in this process.

## 5. The Role of Microbiota in the Occurrence of Synucleinopathies

Alpha-synuclein is an amyloidogenic protein whose pathological conversion is associated with the onset and development of various synucleinopathies. Synucleinopathies include Parkinson's disease, dementia with Lewy bodies, variant Alzheimer's disease with Lewy bodies, and multiple system atrophy. All of these diseases are characterized by the presence of amyloid inclusions in neurons, the main component of which is α-synuclein. Alpha-synuclein is a small (14,460 Da) naturally occurring intrinsically disordered protein that is expressed predominantly in neurons. Its molecule (140 amino acid residues) consists of an amphipathic N-terminus, a hydrophobic middle region called the non-amyloid component, which forms the core of amyloid fibrils, and an acidic C-terminus [81–83].

A large number of in vitro studies have been devoted to determining the molecular mechanisms leading to the pathological aggregation of alpha-synuclein and the factors that influence this process. In the cell, alpha-synuclein exists in two forms: a membrane-bound form enriched in α-helical regions and an unstructured cytosolic form (Figure 3). Alpha-synuclein is converted from monomeric form to fibrillar aggregates via intermediates that are oligomers and protofibrils structurally enriched in β-sheets [84–86]. Fibrillation of alpha-synuclein involves several sequential structural rearrangements that are not yet fully understood [87].

The toxicity of alpha-synuclein has been shown to be due to the presence of oligomeric intermediates [88,89], and the deposition of aggregates of fibrillar alpha-synuclein in the form of Lewy bodies is a neuronal defense mechanism to reduce the toxicity of the oligomeric forms. Oligomers are thought to underlie the molecular basis of synucleinopathies. Thus, cellular processes that lead to the formation of dimers and/or oligomers or that reduce the clearance of these species may be related to alpha-synuclein toxicity [90]. The shape, size, and structure of the resulting fibrils are influenced by a variety of factors. For example, A30P or A53T substitutions in the alpha-synuclein sequence affect the morphology and size of the protofibrils formed. The A30P mutation induces the formation of circular, pore-like protofibrils, whereas A53T induces circular and tubular structures. The wild-type protein also forms ring-like protofibrils after prolonged incubation. The formation of pore-like oligomeric structures helps to explain membrane permeabilization in the presence of alpha-synuclein [91].

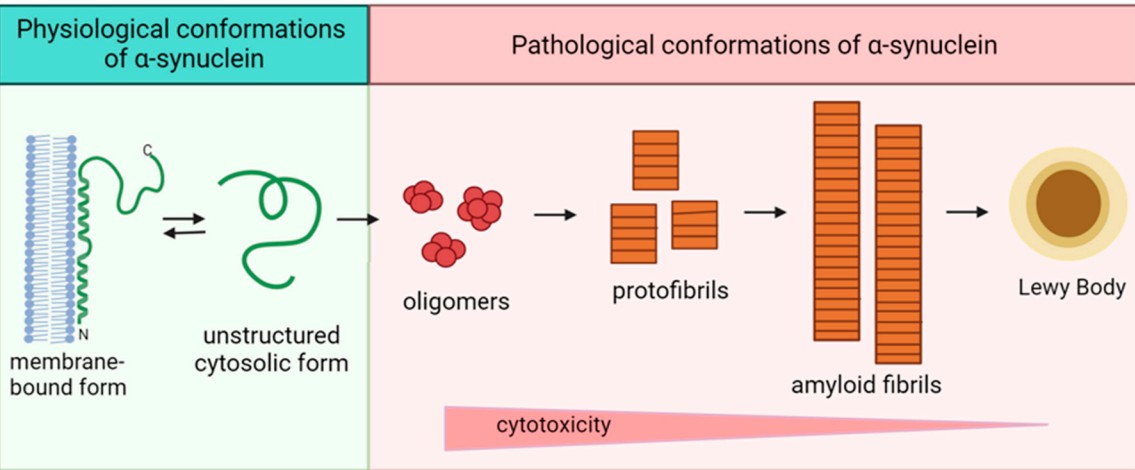

**Figure 3.** Scheme of the pathological transformation of alpha-synuclein. Double arrow is dedicated to the reciprocal transfer of alpha-synuclein forms, single arrows—to the irreversible change of structure.

To understand the relationship between the microbiota and synucleinopathy development, it is necessary to consider a variety of factors that may induce alpha-synuclein amyloid conversion. The formation and stability of oligomers and their subsequent fibrillation can be influenced not only by the mutations mentioned above but also by post-translational modifications of the protein and its interaction with other proteins and compounds present in the body, including those produced by gastrointestinal microorganisms. Compounds formed in the GIT could induce pathological aggregation of alpha-synuclein by altering the structure of its monomers, leading to the exposure of the protein motifs responsible for the association of polypeptide chains with each other. For example, the interaction of alpha-synuclein with phospholipids and polyunsaturated fatty acids in the membrane is known to affect its oligomerization [92,93]. In addition, post-translational modifications of alpha-synuclein induced by intestinal metabolites also affect the aggregation process of this protein. One of the most common protein modifications is the oxidation of cysteine, methionine, and, to a lesser extent, tyrosine residues. Since there are no cysteine residues in the alpha-synuclein sequence, the methionine and tyrosine residues are most susceptible to oxidation. Alpha-synuclein methionine is readily oxidized to methionine sulfoxide by several agents that occur naturally in biological systems. These include hydrogen peroxide, hypochlorite, chloramines, and peroxynitrite [94]. Under physiological conditions, the oxidation of methionine to methionine sulfoxide is mainly reversible. However, there are examples of irreversible methionine oxidation to methioninesulfone under stressful conditions [95]. Incubation of human alpha-synuclein with high concentrations of peroxide leads to the oxidation of all four methionine residues to methionine sulfoxide, whereas at a neutral pH, oxidation does not affect the structure; it remains disordered. At an acidic pH, both proteins (oxidized and unmodified) partially fold. Fibrillation of alpha-synuclein has been shown to be completely inhibited at neutral pH by oxidation of methionine residues, but inhibition disappears at lower pH. The addition of alpha-synuclein with oxidized methionines to wild-type alpha-synuclein inhibits the fibrillation process of the non-oxidized protein [96]. Other post-translational modifications of alpha-synuclein that may be associated with the microbiota include the glycation of alpha-synuclein. Non-enzymatic glycosylation (or glycation) involves modifying the lysine and arginine residues of alpha-synuclein. Glycation of alpha-synuclein has been shown to be one of the most important factors leading to protein aggregation and the formation of Lewy bodies in Parkinson's disease [97,98]. In addition, the glycation of alpha-synuclein by methylglyoxal affects its ability to interact with partner proteins [99]. Thus, a variety of metabolites produced by GIT microorganisms can and should influence the processes of pathological conversion of alpha-synuclein, especially in cells in close proximity to this organ.

There is quite a lot of information on the relationship between synucleinopathies and the microbiota, mainly based on the identification of correlations between these diseases and the presence of certain metabolites in the gut, the composition of which depends on the microbiome [100,101]. Such metabolites, as well as toxic components in food, could stimulate the amyloid transformation of alpha-synuclein upon penetration into neural tissues.

Recent studies have shown that the composition of the microbiota in patients differs from that in healthy individuals [102,103]. In particular, the number of microbial genera that produce short-chain fatty acids and harmful metabolites is reduced in Parkinson's disease [104,105]. It is likely that microbial metabolites may influence disease progression (see Figure 2). In particular, treatment with certain microbial metabolites improved motor function in mouse models of Parkinson's disease [106,107]. Furthermore, in some cases, transplantation of fecal microbiota has been used to prevent the development of Parkinson's disease in mouse models [108]. On the other hand, the transfer of microbiota from Parkinson's patients to healthy mice improves motor dysfunction [109].

However, more specific mechanisms of GIT microbial influence on synucleopathy are virtually unknown. Recently, though, information has emerged on the prion-like mechanism of alpha-synuclein formation in amyloid structures. These observations suggest that the mechanisms previously discovered for prion protein conversion may also apply to alpha-synuclein. The most obvious pathway by which the microbiota influences alpha-synuclein is the induction of its pathological conversion by gut metabolites. It has been established that alpha-synuclein in endocrine cells of the intestinal wall could be altered by intestinal metabolites. As a result, alpha-synuclein could not only adopt a pathological conformation but also penetrate into the tissues of the central nervous system and change the conformation of alpha-synuclein molecules there. Thus, altered forms of alpha-synuclein have been shown to have a prion-like activity, which may be one of the mechanisms involved in the development of this pathology [110,111]. This mechanism may be one of the ways in which synucleinopathies could be induced by metabolites and other components of the intestinal microflora [110].

Although there is a lack of data on the transmission of Parkinson's disease by a prion-like mechanism, it is possible that native or amyloid forms of alpha-synuclein, which enter the gastrointestinal tract through food, could be involved in this type of transmission. Fibrillar forms of alpha-synuclein introduced into the intestinal wall have been shown to be transported from the intestine to the brain and may be involved in the formation of Lewy bodies [112]. Such observations, as well as the above-mentioned data on the spread of the amyloid forms of alpha-synuclein from the enteroendocrine cells along the nerve fibers from the intestine to other tissues, strongly suggest the likelihood of such a mechanism. For sure, this mechanism is more likely to occur if animals' fodder contains nutrients derived from members of the same species, as there is no data on the interspecific transmission of alpha-synuclein infectious particles. In several studies, bacterial and phage chaperones have been shown to be involved in the conversion of alpha-synuclein to its amyloid forms. Several molecular chaperones, such as Hsp27, Hsp60, and Hsp70, have been reported to colocalize with $\alpha$-synuclein in Lewy bodies [113], suggesting that chaperones may play a role in the progression of Parkinson's disease [114]. The effect of two GroEL-like bacteriophage chaperonins, the double-ring EL and the single-ring OBP, on alpha-synuclein fibrillation was investigated in vitro. Regardless of their morphology, both chaperonins accelerated the formation of long amyloid fibrils from alpha-synuclein monomers in an ATP-dependent manner. It seems that OBP is more effective than EL. In contrast, both chaperonins prevented alpha-synuclein amyloid conversion in the absence of ATP [115]. The effect of GroEL on alpha-synuclein transformation has not yet been studied in detail. Currently, it is only confirmed that this chaperonin and its apical domain are able to bind to the amyloid protein in the absence of ATP, thereby slowing down the growth rate of aggregates [116,117].

Consequently, chaperonins in a certain functional state may not prevent the amyloid transformation of alpha-synuclein, as previously assumed, but rather stimulate it. This suggests that bacterial and phage chaperonins may be important players in the development of synucleinopathies. It should be noted that only recently has attention been paid to the relationship between neurodegenerative diseases and viral infections [118].

## 6. Conclusions

The relationship between microbiota composition and the development of neurodegenerative diseases, including those of amyloid nature, is currently unquestionable. However, the specific mechanisms of this relationship are virtually unknown, except for a few examples described in this review. In the case of prion diseases, it is reasonable to assume that bacterial and phage chaperonins of intestinal microorganisms contribute to the pathological transformation of prion protein. The production of specific metabolites by gastrointestinal microorganisms has been shown to induce amyloid forms of alpha-synuclein in intestinal cells, thereby proving to be a contributory factor in synucleinopathies. Obviously, the two described mechanisms may be characteristic of all types of amyloidogenic proteins. Thus, intestinal metabolites may induce pathological transformation of prion protein localized on the surface of nerve cells, and chaperonins circulating in the GIT may contribute to the production of alpha-synuclein amyloid forms. Similar mechanisms may underlie the transformation of other proteins and peptides of amyloid nature, primarily beta-amyloid peptides. The influence of certain microbiota metabolites on the amyloid conversion of prion protein in nerve cells appears to be the most promising area of research. In addition, it is important to elucidate the possible prion-like action of alpha-synuclein entering the gastrointestinal tract in the development of synucleinopathies, primarily Parkinson's disease, and the role of bacterial and phage chaperonins in this process.

**Author Contributions:** Conceptualization, V.I.M. and L.P.K.; methodology, S.S.K.; software, V.I.M.; validation, S.S.K., L.P.K. and E.V.L.; formal analysis, V.I.M.; investigation, S.S.K. and E.V.L.; resources, V.I.M.; data curation, S.S.K. and E.V.L.; writing—original draft preparation, S.S.K. and V.I.M.; writing—review and editing, L.P.K. and E.V.L.; visualization, E.V.L. and S.S.K.; supervision, L.P.K.; project administration, V.I.M.; funding acquisition, L.P.K. All authors have read and agreed to the published version of the manuscript.

**Funding:** The work was supported by the Russian Science Foundation (project No. 23-74-00021).

**Institutional Review Board Statement:** Not applicable.

**Informed Consent Statement:** Not applicable.

**Data Availability Statement:** All data were taken from publicly available articles.

**Conflicts of Interest:** The authors declare no conflict of interest.

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
