# Peer review of "Are Gastrointestinal Microorganisms Involved in the Onset and Development of Amyloid Neurodegenerative Diseases?"

_2036-7481, doi:10.3390/microbiolres14040131_

Round 1
Reviewer 1 Report
Comments and Suggestions for Authors
This review by Muronets et al. takes a fresh look at the influence of the environment on the onset of neurodegenerative disease. The authors summarize the literature on the effect of the state of the microbiota in the spread of the toxic form of the prion protein and the incidence of Parkinson's disease.
Only one question, how can we explain the effect of synuclein in the diet, given that there is a species barrier (does not come from a human source, could animal meal be involved?)?
It is difficult to explain how a-synuclein from food may induce aggregation of the neuronal cellular protein. The toxic effect of food on Parkinson's disease has also been explain by the transit of poisoning chemicals to blood and brain.
Author Response
Dear Reviewer,
We appreciate the time and effort that you dedicated to providing feedback on our manuscript and are grateful for the valuable comments on our paper. We have incorporated most of the suggestions, those changes are highlighted within the manuscript.
Please see the attachent where we have provided point-by-point response to your comments and concerns.

Reviewer 2 Report
Comments and Suggestions for Authors
The manuscript of Muronetz et al. covers an interesting and important topic. However, there are some problems presented below. Firstly, many results are considered that have a rather distant relation to the problem in question. This is particularly the case in Chapter 3, which thus to some extent distracts from the main theme. Shortening this chapter would make it easier to understand the whole text. Chapter 4 is directly related to the key question, but little of the considered data is really clear and interesting. This part would benefit greatly if it includes the most interesting analogous data for other neurodegenerative diseases, especially Alzheimer's, for which GIT involvement has been described. I also could not find a clear explanation as to why bacterial chaperones can play a more important role in amyloid formation than human chaperones. There are also a number of specific points listed below.
Overall, I recommend that the manuscript be resubmitted after substantial rewriting.
Particular issues.
Lines 44-51 and Fig.1: The description of PrPSc is outdated and imprecise. The citations here are also old. In particular, now, that the high resolution spatial structure of PrPSc is established, it sounds strange that "the most important" about PrPSc is just "more β-sheets and pathological aggregation". Essential characteristics of PrPSc are that it forms fibers based on regular cross-beta structure, where PrP molecules are aligned to each other parallel in-register. See, for example: Pathogenic prion structures at high resolution, 2022, PMID 35771767.
It may be worth to stress a paradox that the prion is formed by the globular С domain, rather than unstructured N domain.
Figure 1 is uninformative and misleading. What is the meaning of "amyloid fibril", where one fibril seems to replicate from another in the way it never occurs. I would suggest simply to remove the "amyloid fibril", since it is based on the same structure as PrPSc, just a longer fibril.
Line 61-66: differing properties of the Kuru, BSE and scrapie are in a strange way united by the words "the SAME applies…"
Ln 77: Simultaneously with what? All the sentence needs rewriting. "including prion protein" -what other proteins are meant here?
Ln 84: developing >> the development of
Ln 96: "accumulation of mutant or infectious 96 forms of PrP disrupts(?) Grp78.." Maybe, sequesters?
Ln 119: "which are in principle unable to adopt the native conformation" – better delete this or at least rephrase
Ln 121: "This effect has been shown" – citation is missing.
Ln 135: "spherical amyloid structures of relatively small size, about 200 nm" – I presume that reviewing others work should be critical. An amyloid sphere of about tenfold of average amyloid diameter is a nonsense. Or are these micelles made of amyloids? Have anyone observed structures of such kind, apart from the authors (Ref 53).
Ln 243: Here, I would not resist mentioning the irony that amyloid forming NAC stands for "non-amyloid component"
Fig.3: numerous questions. All established structures of synuclein fibril are parallel in register. In Fig 3 they look like antiparallel. Two amyloid fibrils are depicted as "amyloid fibril". What is meant here? If it is a fiber composed of two strands, then it should be evident that these are two intertwined fibrils. Also, the peptide strand seems to be uninterrupted along the fibril. "oligomer": In my opinion, there is no thinkable way for the depicted structure to convert into "protofibril" and the structure, as it drawn, can only be an off-pathway product. The references 83-85 could prove, or try to prove that I am wrong, so I can only note that the latest of these references is dated 2006, the time when understanding of the amyloid and related structures was rather approximate.
Comments on the Quality of English LanguageSome improvement is required
Author Response

(The authors gave the same response as above.)

Round 2
Reviewer 2 Report
Comments and Suggestions for Authors
The manuscript has been greatly improved by the introduced corrections and is now almost ready for publication.
I would like to note just one point. In Figure 1, PrPSc differs from PrPC by only one small element, whereas recent data suggest that the rearrangement into the beta structure takes up most of the length of PrP. I advice to account for this and make the Fig. 1 more realistic in this point.
Comments on the Quality of English LanguageWhile the overall quality of the English is acceptable, minor problems can be detected. I would advise the authors to check the language using online tools such as Deepl Write.